# mTOR Inhibitor Treatment in Patients with Tuberous Sclerosis Complex Is Associated with Specific Changes in microRNA Serum Profile

**DOI:** 10.3390/jcm11123395

**Published:** 2022-06-13

**Authors:** Bartłomiej Pawlik, Urszula Smyczyńska, Szymon Grabia, Wojciech Fendler, Izabela Dróżdż, Katarzyna Bąbol-Pokora, Katarzyna Kotulska, Sergiusz Jóźwiak, Julita Borkowska, Wojciech Młynarski, Joanna Trelińska

**Affiliations:** 1Department of Pediatrics, Oncology and Hematology, Medical University of Lodz, ul. Sporna 36/50, 91-738 Lodz, Poland; bartlomiej.pawlik@umed.lodz.pl (B.P.); katarzyna.babol-pokora@umed.lodz.pl (K.B.-P.); wojciech.mlynarski@umed.lodz.pl (W.M.); 2Postgraduate School of Molecular Medicine, Medical University of Warsaw, 02-091 Warsaw, Poland; 3Department of Biostatistics and Translational Medicine, Medical University of Lodz, ul. Mazowiecka 15, 92-215 Lodz, Poland; urszula.smyczynska@umed.lodz.pl (U.S.); szymon.grabia@umed.lodz.pl (S.G.); wojciech.fendler@umed.lodz.pl (W.F.); 4Department of Clinical and Laboratory Genetics, Medical University of Lodz, 92-231 Lodz, Poland; izabela.drozdz@umed.lodz.pl; 5Department of Neurology & Epileptology and Pediatric Rehabilitation, The Children’s Memorial Health Institute, ul. Dzieci Polskich 20, 00-999 Warsaw, Poland; k.kotulska@ipczd.pl (K.K.); j.borkowska@ipczd.pl (J.B.); 6Department of Child Neurology, Medical University of Warsaw, ul. Banacha 1A, 02-097 Warsaw, Poland; sergiusz.jozwiak@wum.edu.pl

**Keywords:** tuberous sclerosis, mTOR inhibitor, sirolimus, microRNA

## Abstract

The aim of this study was to determine the serum profiles of miRNAs in patients with tuberous sclerosis (TSC) upon sirolimus treatment and compare them with those previously treated with everolimus in a similarly designed experiment. Serum microRNA profiling was performed in ten TSC patients before sirolimus therapy and again after 3–6 months using qPCR panels (Exiqon). Of 752 tested miRNAs, 28 showed significant differences in expression between TSC patients before and after sirolimus treatment. Of these, 11 miRNAs were dysregulated in the same directions as in the sirolimus groupcompared with the previously described everolimus group, miR-142-3p, miR-29c-3p, miR-150-5p, miR-425-5p, miR-376a-3p, miR-376a-3p, miR-532-3p, and miR-136-5p were upregulated, while miR-15b-3p, miR-100-5p, and miR-185-5p were downregulated. The most significant changes of expression, with fold changes exceeding 1.25 for both treatments, were noted for miR-136-5p, miR-376a-3p, and miR-150-5p. The results of a pathway analysis of the possible target genes for these miRNAs indicated the involvement of the Ras and MAPK signaling pathway. Upregulation of miR-136, miR-376a-3p, and miR-150-5p was noted in TSC patients treated with mTOR inhibitors, indicating a role in the downregulation of the mTOR pathway. Further studies are needed to determine the relationship between upregulated microRNAs and treatment efficacy.

## 1. Introduction

Tuberous sclerosis complex (TSC) is an autosomal dominant genetic disorder caused by a mutation in either one of the two tumor suppressor genes: *TSC1* at 9q34, encoding for hamartin, or *TSC2* at 16p13.3, encoding for tuberin [1]. Inactivation of one of the TSC genes results in hyperactivation of the mammalian target of rapamycin (mTOR) pathway and consequent abnormalities in many cellular processes, including cellular growth, proliferation, protein synthesis, and metabolic control [2]. The most severe clinical manifestation of TSC comprises various severe neurological disorders, such as epilepsy, autism, mental retardation, and subependymal giant cell astrocytoma (SEGA) of the brain, as well as kidney involvement, including renal angiomyolipomas (AML) or renal cysts; it can also involve pulmonary lymphangioleiomyomatosis (LAM) and cardiac rhabdomyomas [3]. Multiple studies have now confirmed that mTOR inhibitor treatment (everolimus and rapamycin) yields clinical benefits for TSC tumors occurring in the kidney, brain, and lungs. Thus, mTOR inhibitors have become an increasingly used and effective treatment option for patients with TSC [4,5,6].

MicroRNA (miRNA) are small RNA molecules (around 22 nucleotides) that regulate gene expression at the posttranscriptional level, either by translation repression or mRNA cleavage [7]. Various studies have shown that miRNAs affect the mTOR pathway in several ways, either by directly interacting with mTOR itself or by targeting key genes within the pathway, which ultimately affect mTOR function. These genes include upstream regulators of mTOR, such as IGF-R, PI3K, and Akt, and negative regulators, such as PTEN [8].

Although the miRNA profile in serum was dysregulated in TSC patients, which was partially resolved after everolimus treatment [9], few studies have been performed on the expression profiles of the microRNAs present in the serum of patients with TSC.

Therefore, the aim of the present study was to determine the effect of sirolimus treatment on the serum miRNA profile in patients with TSC, and compare the findings with those previously obtained for everolimus treatment in a similarly designed experiment.

## 2. Materials and Methods

### 2.1. Patients

The study group consisted of 10 children with a clinical diagnosis of TSC, recruited at the Department of Neurology and Epileptology, Children’s Memorial Health Institute in Warsaw, between December 2016 and November 2017. Diagnosis was made according to the TSC Consensus Conference Diagnostic Criteria [3]. In nine children, the diagnosis was confirmed with genetic tests. One patient was negative for both *TSC1* and *TSC2* gene mutation, but fulfilled the clinical criteria for TSC diagnosis. The inclusion criteria comprised a clinical diagnosis of TSC and SEGA or AML as an indication for sirolimus therapy. The starting dose of sirolimus was 0.5 mg/m^2^/day to achieve a serum drug concentration of 5–10 ng/mL. Patients were evaluated at two time-points—before the initiation of treatment with sirolimus and three to six months after its start—to perform serum miRNA profiling. The results of miRNA profiling were compared with those of the previously described study of 10 patients with TSC treated with everolimus [9]. The study was conducted in accordance with the Declaration of Helsinki and the study protocol was approved by the Bioethics Committee of the Medical University of Lodz (# RNN/306/13/KE). All individuals, or their legal representatives, gave their written informed consent to take part.

Additionally, raw data collected in previous studies performed in children treated with everolimus were reanalyzed, using the same data mining approach as proposed in the current study. The everolimus group consisted of 10 patients with clinical diagnoses of TSC and SEGA, treated with everolimus. The starting dose of everolimus in the previously described study was 4.5 mg/m^2^/day to maintain a serum drug concentration of 5 to 15 ng/mL. Serum samples for miRNA profiling were obtained at two time points: before the initiation of treatment and again after three months [9].

### 2.2. Molecular Methods

For the identification of *TSC1* and *TSC2* gene mutation, DNA was extracted from the blood samples of the patients using a QIAamp DNA Blood Mini Kit (Qiagen, Hilden, Germany), following the manufacturer’s instructions. DNA samples were normalized to a final 5 ng/µL. A Trusight One sequencing kit (Illumina, San Diego, CA, USA) was used to perform the enrichment and final analyses of the *TSC1* and *TSC2* genes. Each procedure was realized following the manufacturer’s instructions. Sanger DNA sequencing was used for the validation of the identified genetic variants. A detailed protocol for the genetic testing was given elsewhere [10].

Serum samples were obtained from patients with TSC using standard vials with a coagulation-activating agent (Becton-Dickinson, Franklin Lakes, NJ, USA). After clot formation, samples were centrifuged at 2000 rpm for 20 min. Afterward, the serum was collected into standard 0.6 mL Eppendorf vials and stored at −80 °C until testing. A miRCURY™ RNA Isolation Kit-Biofluids (Exiqon, Copenhagen, Denmark) was used for miRNA isolation, according to the manufacturer’s protocol. Quantitative reverse-transcription PCR of 752 different miRNAs was performed using a miRCURY LNA™ Universal RT microRNA PCR kit with ExiLENT SYBR Green, according to the manufacturer’s instructions (Exiqon, Vedbaek, Denmark). Hemolysis was assessed using the miR-451/miR-23a ratio [11]. As a negative result was obtained for all of the samples, profiling of the whole dataset could proceed. Exiqon’s serum panels A and B were used for the profiling of circulating microRNAs.

### 2.3. Statistical Analysis

The miRNA level was normalized based on the mean expression of three miRNAs (let-7a-5p, miR-19a-3p, and miR-20a-5p); these were identified by the reference gene finder NormiRazor [12] as an optimal combination for aggregating current data with previous results on the effect of everolimus treatment in TSC patients [9]. The formula for normalization was:dCq = mean Cq of reference miRNAs (N = 20)-assay Cq (sample).

Higher dCq values thus indicated a higher expression of a given miRNA. Cq values for specific miRNAs higher than 37 were filtered as absent calls. Only the miRNA present in at least half of the sample pairs (before and during sirolimus treatment) were considered for analysis. Pretreatment dCq values were compared with post-treatment values by the paired Student’s *t*-test, and the Benjamini false discovery rate (FDR) correction was applied to detect changes in miRNA expression during treatment. *p*-values below 0.05 and (where applicable) FDR < 0.05 were considered as statistically significant.

Following this, the results for sirolimus treatment were compared with those previously obtained for everolimus treatment in a similarly designed experiment. To avoid bias related to differences in statistical methods, the everolimus data were analyzed again with the same methods and reference miRNA as for the new sirolimus data set.

miRNA target prediction and gene set enrichment analysis for miRNA with expression change by both treatments were performed using miRWalk 3 [13].

## 3. Results

The clinical and genetic characteristics of the patients with TSC are presented in Table 1.

Out of 752 tested microRNAs commonly detected in human serum, 531 were detected in at least one sample. Of that number, 236 were present in at least five pairs (on both timepoints).

From the previous study, the miRNA profiles altered due to everolimus treatment were distinctive for patients with TSC and allowed for discrimination from healthy individuals.

An initial visualization of the obtained data based on UMAP representation of miRNA expression, before and during sirolimus treatment, revealed moderate discrimination of time points (Figure 1a). A paired comparison of miRNA expression before and after sirolimus treatment is presented in Figure 1b. Significant differences in the expression of the 28 miRNAs were observed between the readings taken before and after treatment (Figure 1c). Raw data from the miRNA profiling are presented in Appendix A.

A comparison with the impact of everolimus on miRNA expression identified a number of miRNAs that were significantly dysregulated after treatment with either sirolimus or everolimus. A Venn diagram showing the overlap of miRNA dysregulations between the two mTOR inhibitors is presented in Figure 2a. Eleven miRNAs were found to be dysregulated in the same directions for both inhibitors: eight were upregulated (miR-142-3p, miR-29c-3p, miR-150-5p, miR-425-5p, miR-376a-3p, miR-376a-3p, miR-532-3p, and miR-136-5p), while three were downregulated (miR-15b-3p, miR-100-5p, and miR-185-5p) (Figure 2b). The most significant changes in expression were noted for three microRNAs; miR-136-5p, miR-376a-3p, and miR-150-5p, which demonstrated fold changes exceeding 1.25 for both treatments (Figure 3a–c).

The full comparison of the results of miRNA, both before and after sirolimus treatment, are presented in Appendix A, and the full comparison of results for the everolimus group is in Appendix A.

The results of miRWalk database analysis identified 10 common target genes for these three microRNAs (miR-136-5p, miR-376a-3p, and miR-150-5p), and these are given in Table 2. The Gene Set Enrichment Analysis (GSEA) lists the Kyoto Encyclopedia of Genes and Genomes (KEGG) pathways significantly related to identified target genes (Table 3), and these are presented in Appendix A. Of these, the most significant were the MAPK and Ras signaling pathways.

## 4. Discussion

TSC is a disease characterized by a considerably altered serum miRNA profile. In addition, these alterations seem to be mTOR-dependent, as they were partially reversed by treatment with an mTOR inhibitor, everolimus [9]. To confirm these observations, we examined the effects of a different mTOR inhibitor, sirolimus, on the serum miRNA profile in patients with TSC. After comparison with the previous study, 11 miRNAs were found to be dysregulated in the same directions following both treatments, with the most significant changes in expression being observed for miR-136-5p, miR-376a-3p, and miR-150-5p. This indicated that both mTOR inhibitors may have homogenous effects on the mTOR pathway while acknowledging a link between particular miRNA and mTOR signaling.

The observed trend in miR-136-5p expression changes following sirolimus treatment resembles that observed for studies on microRNA expression profile in the tuberous sclerosis complex cell line TSC^−/−^, in which miR-136-5p was found to be upregulated [14]. This indicates that the expression profile of cellular miRNAs may correspond to their serum level. miR-136-5p seems to also be involved in brain tissue protection, as it was shown to block ischemia-induced brain injury and modulate the inflammatory response in neuronal diseases [15].

The second microRNA significantly dysregulated in our study was miR-150-5p. Although no evidence to date links miR-150-5p with TSC, numerous studies have reported the microRNA to be dysregulated in brain tissue upon infections or traumatic brain injury [16,17,18,19]. As severe neurological disorders including epilepsy, autism, mental retardation, and SEGA of the brain are the most common clinical manifestations of TSC, dysregulation of miR-150-5p in TSC patients may be connected with those features. Taken together, miR-136-5p and miR-150-5p seem to be connected to the mTOR pathway, especially its branches concerning central nervous system pathology.

The third selected microRNA was miR-376a-3p, which was significantly upregulated upon mTOR inhibitor treatment. Similarly, increased expression was noted in a study on liver cancer cells treated with sirolimus and everolimus [20]. miR-376a-3p and the three other miRNAs dysregulated in our study (miR-136-5p, miR-150-5p, and miR-142-3p) were found to be influenced in mouse brains during prion disease: some were elevated during preclinical disease and others at a later stage [16].

While miR-142-3p was significantly dysregulated in the everolimus study, no significant change in expression was noted in the present study. However, it was found to demonstrate a connection with the mTOR pathway; treatment of melanoma cells with the flavonoid Licochalcone A resulted in elevated miR-142-3p expression, followed by the inhibition of the target gene *Rheb* and the suppression of the mTOR signaling pathway [21]. Furthermore, miR-142-3p was found to be significantly increased in epileptogenic tubers of TSC patients, initiating a neuroinflammatory cascade [22]. Another report indicated that miR-142-3p and miR-142-5p were significantly decreased in the mitochondrial fraction and increased in the cytoplasmic fraction at early time points following traumatic brain injury [23].

Treatment with mTOR inhibitors is effective in many manifestations of TSC, such as SEGA tumors, kidney AML, LAM, and epilepsy; however, usually only a partial response is observed [4,5,6,24]. This may indicate that hamartin and tuberin act not only as a complex, but that they may work independently from the mTOR pathway in some ways, or that there may be other factors involved in the process. One explanation is that other signaling pathways are connected to mTOR by the microRNA network. Our present bioinformatic analysis showed the MAPK and Ras signaling pathway to have the most predictive value for the three selected miRNAs. The MAPK/ERK pathway functions as the major effector of the Ras oncoprotein and is a major signaling cascade associated with mTOR, linked by negative feedback to mTOR activation [25]. Previous studies demonstrated that dual inhibition of the mTOR and MAPK signaling pathways effectively blocked *Ts^c−/−^* cell proliferation in vitro, and provided a proof-of-concept demonstration that combination therapy targeting both mTOR and MAPK signaling pathways may have the potential to become a novel strategy for the treatment of TSC patients [26,27]. A recent study found that treatment with the MEK inhibitor significantly reduced seizure activity in TSC mouse models, giving a potential alternative strategy to treat seizures in TSC [28]. Finally, ERK activation was observed in SEGA tissue, suggesting that the MAPK/ERK pathway may be used as a target for treatment for TSC patients with SEGAs, perhaps in combination with mTORC1 inhibitors [29].

The limitations of our work are mostly linked to the lack of statistical power due to the small number of patients included in the study. However, the addition of data previously obtained for everolimus treatment in a similarly designed experiment doubled the size of the group. Secondly, our investigation of *TSC1/TSC2* differences and their interaction with miRNA level had very low statistical power; hence, they were not included in the final results of the study. The different TSC statuses of the study group could have affected the miRNA profile. However, the statistical methods used in our analysis comparing the paired data of miRNA expression should have eliminated that problem. Lastly, no cellular experiments were performed to determine the source of miRNA production upon mTOR inhibitor administration. This would be particularly interesting in the context of multiorgan presentation of TSC.

## 5. Conclusions

The miRNAs miR-136-5p, miR-376a-3p, and miR-150-5p were found to be upregulated in TSC patients treated with mTOR inhibitors, indicating a specific connection with the downregulation of the mTOR pathway. However, further studies are needed to determine the relationship between upregulated microRNAs and treatment efficacy.

## Figures and Tables

**Figure 1 jcm-11-03395-f001:**
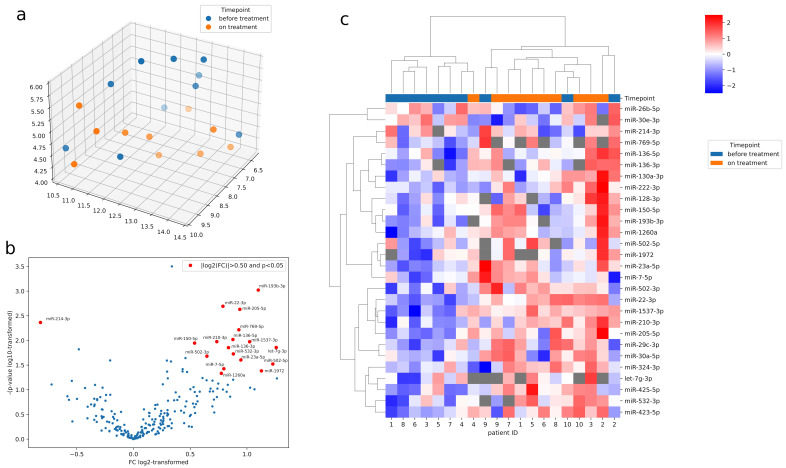
(**a**) UMAP representation of miRNA expression in samples before (blue) and after (orange) sirolimus treatment, missing data imputed with miRNA mean, minimal distance: 0.3, number of neighbors: 4, distance metric: Euclidean, miRNA filtration: expression in at least 5 pairs of samples, and FC calculated on paired samples only. (**b**) Paired comparison of miRNA expression before and after sirolimus treatment. (**c**) Heatmap of 28 significantly dysregulated miRNAs showing significant differences between groups: before (blue) and after (orange) sirolimus treatment (*p* < 0.05 in paired t-test), normalized Cq values scaled in rows, linkage method: Ward, rows (miRNAs) distance: correlation, and columns (samples) distance: Euclidean.

**Figure 2 jcm-11-03395-f002:**
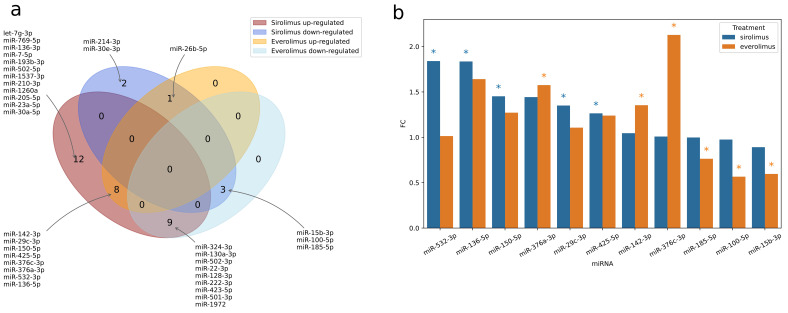
(**a**) Overlap of miRNA dysregulation on sirolimus and everolimus, including miRNAs significantly dysregulated after either of the two treatments. (**b**) miRNAs dysregulated in the same direction by sirolimus and everolimus treatment; asterisks mark significant dysregulation (*p* < 0.05 in paired *t*-test).

**Figure 3 jcm-11-03395-f003:**
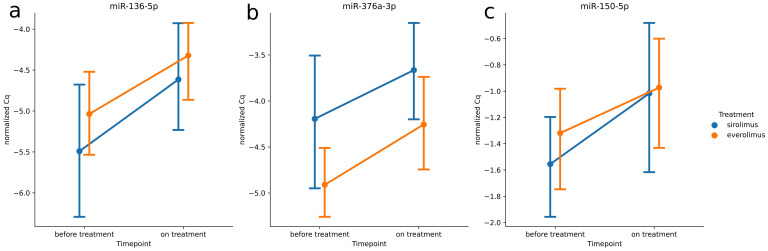
Expression patterns of miRNA expression fold change above 1.25 or below 0.8 after both treatments: (**a**) miR-136-5p, (**b**) miR-376a-3p, and (**c**) miR-150-5p.

**Table 1 jcm-11-03395-t001:** Clinical and genetic characteristics of the study group of patients with TSC.

	Whole Group (N = 10)	TSC2 (N = 7)	TSC1 (N = 2)	No Mutation (N = 1)
Sex (M/F)	5/5	6/1	1/1	1
Age (years)	10.4 ± 4.3	10.3 ± 4.8	8.6 ± 3.3	14.0
SEGA tumor	5/10	3/7	1/2	1
SEN	8/10	5/7	2/2	1
Cortical dysplasia	4/10	3/7	0/2	1
Skin lesions:				
Facial angiofibroma	6/10	4/7	2/2	0
Fibrous cephalic plaque	0/10	0/7	0/2	0
Hypomelanotic macules	7/10	5/7	2/2	0
Shagreen patch	3/10	3/7	0/2	0
Other features:				
Angiomyolipomas	6/10	5/7	1/2	0
Multiple renal cysts	7/10	3/7	0/2	0
Cardiac rhabdomyoma	0/10	0/7	0/2	0
Retinal hamartomas	0/10	0/7	0/2	0
TSC-associated neuropsychiatric disorders	7/10	5/7	2/2	0
Epilepsy	4/10	4/7	0/2	0

**Table 2 jcm-11-03395-t002:** Common target genes of miR-136-5p, miR-150-5p, and miR-376a-3p, and the target prediction algorithms/databases that identified them, in addition to TarPmiR (basic algorithm in miRWalk).

	Target Prediction Method/Database		
Gene	miR-136-5p	miR-150-5p	miR-376a-3p
LONP2	-	-	-
DGKI	TargetScan	TargetScan	-
SMAD2	-	-	-
UQCRB	-	-	-
PCSK5	-	-	miRDB
BDNF	-	-	-
CCNA2	-	-	-
AMOT	-	-	-
P2RY2	TargetScan	-	-
FBXO22	-	-	-

**Table 3 jcm-11-03395-t003:** KEGG pathways involving miR-136-5p, miR-376a-3p, and miR-150-5p targets found by miRWalk, filtered by FDR-adjusted *p*-value below 0.05.

KEGG Pathway Name	*p*-Value	FDR-Adjusted *p*-Value
Ras signaling pathway	0	0
MAPK signaling pathway	0.0006	0.0402
Phospholipase D signaling pathway	0.0005	0.0402
Pathways in cancer	0.0006	0.0402
Rap1 signaling pathway	0.0010	0.0439
Longevity regulating pathway	0.0010	0.0439
Hedgehog signaling pathway	0.0013	0.0439
Neurotrophin signaling pathway	0.0017	0.0439
Glutamatergic synapse	0.0016	0.0439
GABAergic synapse	0.0018	0.0439
Thyroid hormone signaling pathway	0.0017	0.0439
EGFR tyrosine kinase inhibitor resistance	0.0027	0.0441
Longevity regulating pathway	0.0026	0.0441
Cholinergic synapse	0.0021	0.0441
Inflammatory mediator regulation of TRP channels	0.0025	0.0441
Transcriptional misregulation in cancer	0.0022	0.0441
Glioma	0.0028	0.0441
AMPK signaling pathway	0.0032	0.0476

## Data Availability

The data presented in this study are available in Appendix A.

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
