# Peer review of "mTOR Inhibitor Treatment in Patients with Tuberous Sclerosis Complex Is Associated with Specific Changes in microRNA Serum Profile"

_jcm, 2022, doi:10.3390/jcm11123395_

Round 1
Reviewer 1 Report
1. Over all this manuscript is well written.
2. This study conducted mi RNA in SC subjects. Of 752 tested miRNAs, 28 (3.7%) showed significant differences in expression between TSC patients before 23 and after sirolimus treatment.
3. The major problem is the subjects enrolled not all TSC2, but also 2 of TSC 1 and 1 of NMI. The genotype may potentially a bias of the study.
4. Page 1 line 24: ``Of these 11 miRNA were dysregulated in the same directions in both sirolimus and everolimus group`` . Are all the subjects treated with both sirolimis and everolimus? It is confused when compared with the method `` The results of the miRNA 75 profiling were compared to those of a group of 10 patients with TSC previously treated 76 with everolimus`` on page 2 line 75-76.
5. What are the dose of sirolimis and everolimus?
Author Response
Reviewer 1:
- Over all this manuscript is well written.
- This study conducted mi RNA in SC subjects. Of 752 tested miRNAs, 28 (3.7%) showed significant differences in expression between TSC patients before 23 and after sirolimus treatment.
- The major problem is the subjects enrolled not all TSC2, but also 2 of TSC 1 and 1 of NMI. The genotype may potentially a bias of the study.
Response:
We agree that subjects enrolled in the study have different TSC status: seven patients have TSC2 mutation, two patients TSC1, and one – NMI. It could potentially affect the miRNA profile. However statistical methods used in our analysis compared paired data of miRNA expression.
We have added this problem to the limitations of our study.
“The different TSC status of the study group could potentially affect the miRNA profile. However statistical methods used in our analysis comparing paired data of miRNA expression should eliminate that problem.”
- Page 1 line 24: ``Of these 11 miRNA were dysregulated in the same directions in both sirolimus and everolimus group`` . Are all the subjects treated with both sirolimis and everolimus? It is confused when compared with the method `` The results of the miRNA 75 profiling were compared to those of a group of 10 patients with TSC previously treated 76 with everolimus`` on page 2 line 75-76.
Response:
According to reviewer’s suggestion we have change the sentence in abstract (line24) and the sentence in method section – line 75-76, to make it clear that it was two groups of patients. One group previously described treated with everolimus and second group currently studied treated with sirolimus.
“Of these 11 miRNA were dysregulated in the same directions in sirolimus group compared to previously described everolimus group”
“The results of miRNA profiling were compared to those in previously described study of 10 patients with TSC treated with everolimus”
- What are the dose of sirolimis and everolimus?
Response:
The starting dose of sirolimus was 0.5 mg/m2/day to achieve serum drug concentration of 5 -10 ng/ml.
The starting dose of everolimus in previously described study was 4.5 mg/m2/day to maintain through serum drug concentration of 5 to 15 ng/ml.
We have added these data to Materials and Methods section.
Reviewer 2 Report
Palik et al studied the profile of serum miRNA in patients with tuberous sclerosis complex (TSC), 10 treated with sirolimus and 10 with everolimus, and observed changes in the expression of several miRNAs. miRNA changes before treatment suggested the involvement of the Ras and MAPK signaling pathway, and those after treatment were considered to be associated with downregulation of the mTOR pathway.
This study is well designed and conducted. The results are nicely and succinctly described.
Minor issue:
It was difficult for me to interpret the heatmap (Figure 1c). The legend could be improved.
Author Response
Dear Editor,
We greatly appreciate your interest in our manuscript and made our best effort to introduce the changes suggested by the reviewers. The responses to each of their concerns are listed below. We hope that the revised manuscript will prove to be of sufficient quality to consider its publication in your prestigious journal.
Thank you for your consideration of our paper.
We look forward to hearing from you.
With kindest regards,
Joanna Trelinska M.D., Ph.D.
Manuscript ID: jcm-1758907 
Comments and Suggestions for Authors
Reviewer 2:
Palik et al studied the profile of serum miRNA in patients with tuberous sclerosis complex (TSC), 10 treated with sirolimus and 10 with everolimus, and observed changes in the expression of several miRNAs. miRNA changes before treatment suggested the involvement of the Ras and MAPK signaling pathway, and those after treatment were considered to be associated with downregulation of the mTOR pathway.
This study is well designed and conducted. The results are nicely and succinctly described.
Minor issue:
It was difficult for me to interpret the heatmap (Figure 1c). The legend could be improved.
Response:
According to the reviewer’s suggestion we have changed the legend of the Figure 1c to make it more clear for interpretation.
Figure legends:
Figure 1. a) UMAP representation of miRNA expression in samples before (blue) and after (orange) sirolimus treatment, missing data imputed with miRNA mean, minimal distance: 0.3, number of neighbors: 4, distance metric: Euclidean, miRNA filtration: expression in at least 5 pairs of samples, FC calculated on paired samples only. b) Paired comparison of miRNA expression before and after sirolimus treatment. c) Heatmap of 28 significantly dysregulated miRNAs showed significant differences between groups: before (blue) and after (orange) sirolimus treatment (p<0.05 in paired t-test), normalized Cq values scaled in rows, linkage method: Ward, rows (miRNAs) distance: correlation, columns (samples) distance: Euclidean.
This manuscript is a resubmission of an earlier submission. The following is a list of the peer review reports and author responses from that submission.